# Numerical Approximations of Diblock Copolymer Model Using a Modified Leapfrog Time-Marching Scheme

Lizhen Chen [1], Ying Ma [2,*], Bo Ren [1] and Guohui Zhang [1]

[1] Beijing Computational Science Research Center, Beijing 100193, China; lzchen@csrc.ac.cn (L.C.); renbo_mech@csrc.ac.cn (B.R.); zhangguohui@csrc.ac.cn (G.Z.)

[2] Department of Mathematics, Faculty of Science, Beijing University of Technology, Beijing 100124, China

[*] Correspondence: maying@bjut.edu.cn

**Abstract:** An efficient modified leapfrog time-marching scheme for the diblock copolymer model is investigated in this paper. The proposed scheme offers three main advantages. Firstly, it is linear in time, requiring only a linear algebra system to be solved at each time-marching step. This leads to a significant reduction in computational cost compared to other methods. Secondly, the scheme ensures unconditional energy stability, allowing for a large time step to be used without compromising solution stability. Thirdly, the existence and uniqueness of the numerical solution at each time step is rigorously proven, ensuring the reliability and accuracy of the method. A numerical example is also included to demonstrate and validate the proposed algorithm, showing its accuracy and efficiency in practical applications.

**Keywords:** phase field; linear scheme; energy stable; Diblock copolymer



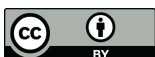

## 1. Introduction

Block copolymers are a unique class of polymers characterized by their structural organization. They consist of two or more chemically distinct monomer blocks that are covalently bonded together within a single polymer molecule. These monomer blocks are arranged in a repetitive pattern, typically in the form of alternating blocks, and are connected by covalent bonds. One of the remarkable features of block copolymers is their ability to undergo self-assembly, leading to the formation of well-defined and ordered nanostructures. This property has garnered significant attention and interest in various scientific and engineering fields, particularly in the realm of nanotechnology applications. Researchers and engineers have recognized the potential of block copolymers to create highly tailored and controlled nanostructures, which can have various applications [1–5].

The governing diblock copolymer model is known to follow an energy dissipation law, but due to its nonlinearity, maintaining unconditional energy stability at the discrete level poses a significant challenge. In fact, both the explicit and simple fully-implicit type discretizations will impose severe time step constraints, depending on the interfacial width (cf. [6–8]). Recently, some efforts have been made to develop first-order schemes which are accurate in time and energy-stable for the diblock copolymer model (cf. [2,9,10]). These schemes are primarily based on either the nonlinear convex splitting approach (cf. [9,11]) or the linear stabilization approach (cf. [12–24]). Obviously, higher order time-marching schemes that preserve the unconditional energy stability are more preferable. To address this challenge, several numerical approaches have been developed for this model, each with its own techniques. Some of the notable approaches include the IEQ (invariant energy quadratization) approach [25] which was used for the double well potential in the context of the diblock copolymer model. It involves a technique known as energy quadratization, which is employed to stabilize the numerical scheme and ensure energy stability. Additionally, the SAV (scalar auxiliary variable) approach [26] is utilized with a stabilization technique to create a novel stabilized SAV method. This method incorporates a crucial linear stabilization term that enhances stability and allows for the use of larger time steps while maintaining the required accuracy. It has been applied not only to the

diblock copolymer model but also extended to the magnetic-coupled Cahn–Hilliard phase-field model for diblock copolymers. Also, the work [27] introduces a new class of linear time-integration schemes for phase-field models, including the diblock copolymer model. These schemes extend the energy quadratization technique by introducing additional free parameters, which serve to further stabilize the schemes and improve their accuracy. These approaches represent the ongoing efforts in developing numerical methods that can handle the nonlinearity and energy stability requirements of the diblock copolymer model. They play a crucial role in enabling simulations and computations related to diblock copolymers, which have applications in various scientific and engineering domains.

In this paper, the main objective is to develop a linear, second-order-accurate time-marching scheme for solving the block copolymer model. This scheme is based on the leapfrog method, which has been recently developed and applied in similar contexts [28,29]. The contributions and advantages of this newly proposed scheme are as follows: Firstly, the newly proposed scheme maintains that unconditional energy stability can be strictly proven. This is a crucial property as it ensures that the numerical solution does not exhibit unphysical behavior and remains stable throughout the simulation. Notably, this stability is expressed in terms of the original variables used in the model. Secondly, unlike some other numerical approaches (e.g., invariant energy quadratization [30] or scalar auxiliary variable approaches [31]), the newly proposed scheme satisfies an energy dissipation law that is expressed in terms of the original variables of the block copolymer model. This ensures the accuracy and consistency of the scheme in preserving the physical properties of the system. Thirdly, the newly proposed scheme is entirely linear in nature. This means that at each time-marching step, only a linear system needs to be solved. This linearity is in contrast to convex-splitting nonlinear schemes [11,32], which typically require the application of nonlinear iterative Newton's methods. The linearity of the scheme can significantly enhance computational efficiency. Fourthly, the existence and uniqueness of the numerical solution can be easily verified within the framework of the newly proposed scheme. This is important for ensuring the reliability and predictability of the numerical results obtained through the scheme. Overall, the paper presents a novel approach to solving the block copolymer model that combines the advantages of unconditional energy stability, conservation of energy dissipation laws, linearity, and ease of verifying the numerical solution's existence and uniqueness. These features make the proposed scheme a promising and effective tool for numerical simulations of block copolymer modeling.

The paper's structure and organization are outlined as follows: The Section 1 provides an introduction to the key topic and presents the main contributions and advantages of the proposed numerical scheme for solving the block copolymer model. A brief introduction to the governing phase field models that are relevant to diblock copolymers is also given. Section 2 presents an overview of the mathematical framework and equations that describe the behavior of diblock copolymers. The details of the proposed numerical scheme are described in Section 3. It explains how the scheme achieves second-order temporal accuracy in temporal discretization and employs the Fourier pseudo-spectral method for spatial discretization. A numerical experiment to validate and demonstrate the accuracy and efficiency of the proposed numerical scheme is presented in Section 4. This experiment involves solving a specific problem or scenario related to block copolymers using the developed scheme. Finally, some concluding remarks are made with a summary of the main contributions.

## 2. Phase Field Models for Diblock Copolymers

The phase field diblock copolymer model is represented by a partial differential equation (PDE) involving the phase field variable $\phi$.

$$\phi_t = -M\left(\varepsilon^2 \Delta^2 \phi - \Delta((\phi^2 - 1)\phi) + \alpha(\phi - \bar{\phi})\right), \tag{1}$$

where $\phi$ is represented as an order parameter representing the local concentration difference of the components, $M > 0$ is the mobility parameter, and $\varepsilon > 0$ and $\alpha > 0$ are physical

parameters. The average concentration of $\phi$ is denoted as $\bar{\phi}$. It is calculated as the integral of $\phi$ over the entire domain divided by the domain's volume: $\bar{\phi} = \frac{1}{|\Omega|} \int_\Omega \phi(x, y) d\Omega$.

For simplicity, we assume a periodic boundary condition for the diblock copolymer model in (1) in the rest of this section. Let

$$\psi = (-\Delta)^{-1}(\phi - \bar{\phi}), \tag{2}$$

then the total phenomenological free energy associated with the phase field diblock copolymer model is defined as follows

$$E(\phi) = \int_\Omega \left( \frac{\varepsilon^2}{2} |\nabla \phi|^2 + \frac{1}{4}(\phi^2 - 1)^2 + \frac{\alpha}{2}|\nabla \psi|^2 \right) d\Omega. \tag{3}$$

In general, the model preserves the mass conservation property

$$\frac{d}{dt} \int_\Omega \phi(\mathbf{x}, t) d\Omega = 0, \tag{4}$$

where $\mathbf{n}$ is the outward normal on the boundary.

Then, we can easily to obtain the energy dissipation in time

$$\frac{dE}{dt} = \int_\Omega \frac{\delta E}{\delta \phi} \frac{\delta \phi}{\delta t} d\Omega = -\int_\Omega M |\nabla \left( \varepsilon^2 \Delta \phi - (\phi^2 - 1)\phi + \alpha \psi \right)|^2 d\Omega. \tag{5}$$

Next, we will give an efficient numerical scheme to the diblock copolymer model in (1).

Due to the fourth-order space derivative of the system (1), it will cause lots of stability trouble in numerical simulation. Therefore, we can rewrite the system (1) to reduce the space derivative order as follows

$$\begin{cases} \phi_t = -M\Delta\mu, \\ \mu = -\varepsilon^2 \Delta\phi - (\phi^2 - 1)\phi + \alpha\psi, \\ \psi = (-\Delta)^{-1}(\phi - \bar{\phi}). \end{cases} \tag{6}$$

Based on the above partial differential equations, we will give the temporal and space discretization.

## 3. Numerical Schemes

In this section, we will introduce a numerical approach that combines the leapfrog method for handling time discretization and the Fourier spectral method for spatial discretization.

### 3.1. Temporal Discretization

Suppose the time domain of the diblock copolymer model in (1) is $[0, T]$. Then we discretize the time domain $[0, T]$ using a uniform time step, $0 = t_0 < t_1 < t_2 < \ldots < t_N = T$, with $t_n = \frac{n}{N}T$, and we denote the numerical solutions with respect to time discretization at $t_n$ as $\phi^n$, $0 \leq n \leq N$. Now, we propose a second-order time-discrete and linearity numerical scheme in time as below.

**Scheme 1.** *After we obtain $\phi^{n-1}$ and $\phi^n$, we can calculate $\phi^{n+1}$ using the following step scheme:*
- *Solve for $\phi^{n+1}$ using the following scheme*

$$\frac{1}{2\Delta t}(\phi^{n+1} - \phi^{n-1}) = M\Delta\mu^n, \tag{7}$$

$$\mu^n = -\varepsilon^2 \Delta \frac{\phi^{n+1} + \phi^{n-1}}{2} + (\phi^n)^2 \frac{\phi^{n+1} + \phi^{n-1}}{2} - \phi^n + \alpha \frac{\psi^{n+1} + \psi^{n-1}}{2}, \tag{8}$$

*The system can have either periodic boundary conditions or the following the physical boundary conditions:*

$$\nabla \phi^{n+1} \cdot \mathbf{n}|_{\partial \Omega} = 0, \quad \nabla \mu^n \cdot \mathbf{n}|_{\partial \Omega} = 0, \tag{9}$$

*where*

$$\psi^{n+1} := (-\Delta)^{-1}(\phi^{n+1} - \bar{\phi}^{n+1}) \text{ and } \psi^{n-1} := (-\Delta)^{-1}(\phi^{n-1} - \bar{\phi}^{n-1}). \tag{10}$$

**Remark 1.** *Given the initial condition $\phi^0$, the first step $\phi^1$ can be obtained in a second-order accurate way in terms of $\Delta t$ via a first-order time-marching scheme, which was implemented using significantly smaller time steps. For example, we update $\phi^1$ as follows*

$$\frac{1}{\Delta t^1}(\phi^1 - \phi^0) = M\Delta \mu^0, \tag{11}$$

$$\mu^0 = -\varepsilon^2 \Delta \phi^1 + (\phi^0)^3 - \phi^0, \tag{12}$$

*with $\Delta t^1 = \frac{\Delta t}{\left[\frac{1}{\Delta t}+1\right]}$, $K = \left[\frac{1}{\Delta t} + 1\right]$.*

**Theorem 1.** *The semi-discrete numerical Scheme 1 possesses the property of unconditional energy stability. The discrete energy law is as follows*

$$E^{n+1,n} - E^{n,n-1} = -\Delta t M(\nabla \mu^n, \nabla \mu^n), \tag{13}$$

*where*

$$
\begin{aligned}
E^{n+1,n} &= \frac{\varepsilon^2}{4}(\|\nabla \phi^{n+1}\|^2 + \|\nabla \phi^n\|^2) + (\frac{1}{4}(\phi^{n+1}\phi^n - 1)^2, 1) + \frac{\alpha}{4}(\|\nabla \psi^{n+1}\|^2 + \|\nabla \psi^n\|^2), \\
E^{n,n-1} &= \frac{\varepsilon^2}{4}(\|\nabla \phi^n\|^2 + \|\nabla \phi^{n-1}\|^2) + (\frac{1}{4}(\phi^n\phi^{n-1} - 1)^2, 1) + \frac{\alpha}{4}(\|\nabla \psi^n\|^2 + \|\nabla \psi^{n-1}\|^2).
\end{aligned}
\tag{14}
$$

**Proof.** Taking the inner product of (7) with $\Delta t \mu^n$, we obtain

$$\left(\frac{\phi^{n+1} - \phi^{n-1}}{2}, \mu^n\right) = -M\Delta t(\nabla \mu^n, \nabla \mu^n). \tag{15}$$

Next, taking the inner product of (8) with $\frac{1}{2}(\phi^{n+1} - \phi^{n-1})$, we have

$$
\begin{aligned}
\left(\frac{\phi^{n+1} - \phi^{n-1}}{2}, \mu^n\right) &= \frac{\varepsilon^2}{4}\left(\|\nabla \phi^{n+1}\|^2 - \|\nabla \phi^{n-1}\|^2\right) + \frac{1}{4}\Big((\phi^{n+1}\phi^n)^2 - (\phi^n\phi^{n-1})^2 \\
&\quad -2\phi^{n+1}\phi^n + 2\phi^n\phi^{n-1}, 1\Big) + \alpha(\frac{\psi^{n+1} + \psi^{n-1}}{2}, \frac{1}{2}(\phi^{n+1} - \phi^{n-1})).
\end{aligned}
\tag{16}
$$

Then, we derive from (10) that

$$-\Delta(\psi^{n+1} - \psi^{n-1}) = \phi^{n+1} - \phi^{n-1} - (\bar{\phi}^{n+1} - \bar{\phi}^{n-1}). \tag{17}$$

By taking the $L^2$ inner product of (17) with $\frac{\alpha}{4}(\psi^{n+1} + \psi^{n-1})$, we obtain

$$
\begin{aligned}
\frac{\alpha}{4}(\|\nabla \psi^{n+1}\|^2 - \|\nabla \psi^{n-1}\|^2) &= \frac{\alpha}{4}(\psi^{n+1} + \psi^{n-1}, \phi^{n+1} - \phi^{n-1}) - \frac{\alpha}{4}(\psi^{n+1} + \psi^{n-1})(\bar{\phi}^{n+1} - \bar{\phi}^{n-1}, 1) \\
&= \frac{\alpha}{4}(\psi^{n+1} + \psi^{n-1}, \phi^{n+1} - \phi^{n-1})
\end{aligned}
\tag{18}
$$

Subtracting Equations (16) and (18) from (15), we have

$$E^{n+1,n} - E^{n,n-1} = -M\Delta t(\nabla \mu^n, \nabla \mu^n). \tag{19}$$

This completes the proof. □

To establish a comparison with the scheme developed above, we also list the traditional second-order convex-splitting scheme based on the Crank–Nicolson formula [26] as follows:

**Scheme 2.** *After obtaining $\phi^n$, we compute $\phi^{n+1}$ by:*

$$
\begin{cases}
\dfrac{\phi^{n+1} - \phi^n}{M\Delta t} = \Delta w^{n+\frac{1}{2}}, \\[2mm]
w^{n+\frac{1}{2}} = -\varepsilon^2 \Delta \phi^{n+\frac{1}{2}} + \left( \dfrac{(\phi^{n+1})^2 + (\phi^n)^2}{2} - 1 \right) \dfrac{\phi^{n+1} + \phi^n}{2} + \alpha \psi^{n+\frac{1}{2}}, \\[2mm]
\psi^{n+\frac{1}{2}} = -(\Delta^{-1}(\phi^{n+\frac{1}{2}} - \bar{\phi}^{n+\frac{1}{2}})),
\end{cases}
\tag{20}
$$

*where $\phi^{n+\frac{1}{2}} = \dfrac{\phi^{n+1} + \phi^n}{2}$ and $\psi^{n+\frac{1}{2}} = \dfrac{\psi^{n+1} + \psi^n}{2}$. The unconditional energy stability of the above scheme can be easily proven by employing the comparable idea presented in [32,33]. This scheme is a nonlinear scheme which needs the Newton iteration or the fixed point iteration method. Nonetheless, our proposed scheme is a fully linear scheme which can be solved directly in each time-marching step.*

*3.2. Spatial Discretization*

In this paper, it is natural to utilize the accurate and efficient Fourier pseudo-spectral method under the periodic boundary conditions. For the sake of simplicity, we will focus on a two-dimensional domain in this study. However, it is worth noting that an extension from a two-dimensional domain to a three-dimensional domain can be achieved using the tensor product method.

Let $L_x$ and $L_y$ denote the lengths in each direction in the spatial domain $\Omega = [0, L_x] \times [0, L_y]$. Let $N_x$ and $N_y$ be two positive even integers. We discretize $\Omega$ using uniformly partitioned meshes with mesh sizes $h_x = L_x/N_x$ and $h_y = L_y/N_y$. Therefore, the discrete domain is given by

$$
\Omega_h = \left\{ (x_i, y_j) \mid x_i = ih_x, y_j = jh_y, 0 \le i \le N_x - 1, 0 \le j \le N_y - 1 \right\}. \tag{21}
$$

Furthermore, we introduce the space of grid functions on $\Omega_h$, denoted as

$$
V_h = \left\{ u \mid u = \{ u_{ij} \mid (x_i, y_j) \in \Omega_h, 0 \le i \le N_x - 1, 0 \le j \le N_y - 1 \}. \right\} \tag{22}
$$

For any functions $F \in V_h, G \in V_h$, we define the induced discrete inner product and $l_2$ norm as follows

$$
(F, G)_h = \sum_{i=1}^{N_x - 1} \sum_{j=1}^{N_y - 1} F_{ij} G_{ij} h_x h_y, \quad \| F \|_h = \sqrt{(F, F)_h}. \tag{23}
$$

The notations for the discrete gradient operator and the discrete Laplace operator are defined as $\nabla_h$ and $\Delta_h$, respectively. To distinguish it from the semi-discrete solution $\phi^n$ ($n \ge 0$), we denote the full discrete solution with a subscript as $\phi_N^n \in V_h$.

The proposed fully discrete numerical scheme is as follows.

**Scheme 3.** *After we obtain the previous numerical solution $\phi_N^{n-1}$, $\phi_N^n \in V_h$, we update $\phi_N^{n+1} \in V_h$, with $n \ge 1$, via following step:*

- *We solve for $\phi_N^{n+1}$ and $\hat{\mathbf{u}}_N^{n+1}$ using the following scheme:*

$$
\frac{1}{2\Delta t}(\phi_N^{n+1} - \phi_N^{n-1}) = M\Delta_h \mu_N^n, \tag{24}
$$

$$\begin{aligned}
\mu_N^n &= -\varepsilon^2 \Delta_h \frac{\phi_N^{n+1} + \phi_N^{n-1}}{2} + (\phi_N^n)^2 \frac{\phi_N^{n+1} + \phi_N^{n-1}}{2} - \phi_N^n \\
&\quad + \alpha \frac{\psi_N^{n+1} + \psi_N^{n-1}}{2},
\end{aligned} \tag{25}$$

with periodic boundary conditions.

Some theoretical results can be obtained for the fully discrete scheme presented in Scheme 3.

**Theorem 2.** *The fully discrete Scheme 3 guarantees unconditional energy stability. Furthermore, the fully discrete numerical solutions satisfy the following energy dissipation law*

$$E_h^{n+1,n} - E_h^{n,n-1} = -\Delta t M \|\nabla_h \mu_N^n\|_h^2, \tag{26}$$

*where*

$$\begin{aligned}
E_h^{n+1,n} &= \frac{\varepsilon^2}{4}(\|\nabla \phi_N^{n+1}\|_h^2 + \|\nabla \phi_N^n\|^2) + \frac{1}{4}((\phi_N^{n+1}\phi_N^n - 1)^2, 1)_h + \frac{\alpha}{4}(\|\nabla \psi_N^{n+1}\|^2 + \|\nabla \psi_N^n\|^2), \\
E_h^{n,n-1} &= \frac{\varepsilon^2}{4}(\|\nabla \phi_N^n\|_h^2 + \|\nabla \phi_N^{n-1}\|_h^2) + \frac{1}{4}((\phi_N^n\phi_N^{n-1} - 1)^2, 1)_h + \frac{\alpha}{4}(\|\nabla \psi_N^n\|^2 + \|\nabla \psi_N^{n-1}\|^2).
\end{aligned} \tag{27}$$

**Proof.** The proof can follow the idea of the time-discrete Scheme 1. Here, we provide a brief proof. Firstly, we take the inner product of (24) with $\Delta t \mu_N^n$. Then, we take the inner product of (25) with $\frac{1}{2}(\phi_N^{n+1} - \phi_N^{n-1})$. By subtracting these two inner products, we obtain

$$E_h^{n+1,n} - E_h^{n,n-1} = -M\Delta t \|\nabla_h \mu_N^n\|_h^2, \tag{28}$$

which completes the proof.   □

**Theorem 3.** *The full discrete Scheme 3 is uniquely solvable, meaning that there exists a unique solution $\phi_N^{n+1} \in V_h$ for each time-marching step.*

**Proof.** We need to demonstrate the existence of a unique solution for each step of the fully discrete Scheme 3.

For the step in Equations (24) and (25), it can be expressed as an algebraic linear system $AX = b$, where $X = (\mu_N^n, \phi_N^{n+1}, \psi_N^{n+1})$ and

$$A = \begin{bmatrix} -M\Delta_h & \frac{1}{2\Delta t} & 0 \\ -1 & -\frac{\varepsilon^2}{2}\Delta_h + \frac{(\phi_N^n)^2}{2} & \frac{\alpha}{2} \\ 0 & -I & -\Delta_h \end{bmatrix}, \tag{29}$$

and $b$ are the remaining explicit terms. To demonstrate that $AX = b$ has a unique solution, it suffices to prove that $AX = \mathbf{0}$ only has the solution $X = \mathbf{0}$.

It is evident that $X = \mathbf{0}$ is a solution to $AX = \mathbf{0}$. Next, we will establish that $X = \mathbf{0}$ if $AX = \mathbf{0}$. Let $X = (X_1, X_2, X_3)$. In fact, if we take the inner product of $Y = (2X_1, \frac{X_2}{\Delta t}, \frac{\alpha}{2\Delta t}X_3)$ on both sides of $AX = 0$, we obtain

$$2\|\sqrt{M}\nabla_h X_1\|_h + \frac{\varepsilon^2}{2}\|\nabla_h X_2\|_h^2 + (\frac{(\phi_N^n)^2}{2}, X_2^2)_h + \frac{\alpha}{2}\|\nabla_h X_3\|_h^2 = 0. \tag{30}$$

We immediately observe that $\|X_2\|_h^2 = 0$, which implies $X_2 = \mathbf{0}$. Using the given fact $(-\Delta_h)^{-1}X_3 = X_2 - \bar{X}_2 = 0$ and the uniqueness of the inverse of the Laplacian operator $\Delta_h$, we can conclude that $X_3 = 0$. From the second equation of $AX = \mathbf{0}$, we obtain

$$X_1 = \frac{\varepsilon^2}{2}\Delta_h X_2 + \left(\frac{(\phi_N^n)^2}{2}\right)X_2 + \frac{\alpha}{2}X_3 = \mathbf{0}. \tag{31}$$

Therefore, we have shown that $X = \mathbf{0}$, provided that $AX = \mathbf{0}$.

This indicates that the step in Equations (24) and (25) has a unique solution. Hence, we have shown that Scheme 3 also has a unique solution when fully discretized. $\square$

**Theorem 4.** *The fully discrete scheme presented in Scheme 3 preserves the total mass of the phase variable, given by*

$$(\phi_N^{n+1}, 1)_h = (\phi_N^n, 1)_h, \quad \forall n \geq 0. \tag{32}$$

**Proof.** To prove the preservation of total mass, we take the discrete inner product of Equation (24) with $\phi_N^{n+1}$ and ensure that the starting values preserve the total mass, i.e., $(\phi_N^1, 1)_h = (\phi_N^0, 1)_h$. $\square$

Similarly, we present the fully discrete scheme of the phase field diblock copolymer model 1 as follows:

**Scheme 4.** *After obtaining $\phi_N^n$, we compute $\phi_N^{n+1}$ by:*

$$\begin{cases} \dfrac{\phi_N^{n+1} - \phi_N^n}{M\Delta t} = \Delta w_N^{n+\frac{1}{2}}, \\[2mm] w_N^{n+\frac{1}{2}} = -\varepsilon^2 \Delta \phi_N^{n+\frac{1}{2}} + \left(\dfrac{(\phi_N^{n+1})^2 + (\phi_N^n)^2}{2} - 1\right)\dfrac{\phi_N^{n+1} + \phi_N^n}{2} + \alpha \psi_N^{n+\frac{1}{2}}, \\[2mm] \psi_N^{n+\frac{1}{2}} = -(\Delta^{-1}(\phi_N^{n+\frac{1}{2}} - \bar{\phi}_N^{n+\frac{1}{2}})), \end{cases} \tag{33}$$

*where $\phi_N^{n+\frac{1}{2}} = \dfrac{\phi_N^{n+1} + \phi_N^n}{2}$ and $\psi_N^{n+\frac{1}{2}} = \dfrac{\psi_N^{n+1} + \psi_N^n}{2}$.*

## 4. Numerical Examples for the Diblock Copolymer Model

The focus of this section is the study of spinodal decomposition, a feature of phase separation dynamics. The example begins by testing the accuracy of the numerical Scheme 3, which is a crucial component of the methodology for simulating and analyzing spinodal decomposition. The accuracy test aims to evaluate how well the numerical scheme approximates the actual behavior of the phase separation process. Such tests are essential to validate the reliability of the numerical method and ensure accurate results. After the accuracy test, the example presents numerical simulations, which involve applying the numerical scheme to specific scenarios or initial conditions related to spinodal decomposition. The purpose of these simulations is to observe and analyze how the phase separation dynamics evolve over time according to the numerical scheme.

### 4.1. Accuracy Test

First of all, we list the setup for the accuracy test of the numerical Scheme 3. The computational domain is defined as $\Omega = [0, 4\pi]^2$. The parameters used in the simulation include mobility $M = 0.02$, phase width $\varepsilon = 0.05$, and nonlocal parameter $\alpha = 5$.

The initial condition for the concentration field $\phi$ at $t = 0$ is specified as a randomly perturbed concentration field, defined as:

$$\phi|_{t=0} = c_0 + \text{rand}(-0.001, 0.001), \tag{34}$$

where $c_0$ is a constant. rand$(-0.001, 0.001)$ generates a random number following a normal distribution in the range $[-0.001, 0.001]$. The spatial domain is discretized using a grid consisting of $256 \times 256$ grid points.

Since the exact solution for the diblock copolymer model is unknown, we take the numerical results of Scheme 3 with $T = 1$ and $\Delta t = 1.0^{-4}$ as the exact solution. We compute the numerical errors at a specific time point, namely at $t = 1$. It then evaluates the $L^2$-errors as a function of the time step $\Delta t$. The errors are calculated for different values of the parameter $\alpha$ while fixing the initial condition. Specifically, the values $\alpha = 0.0$ and $\alpha = 5.0$ are considered. The results presented in Figure 1 show a graphical representation of the $L^2$-errors as a function of the time step $\Delta t$ for both values of the physical parameter $\alpha$. We observe a second-order convergence rate at the time step $\Delta t$. We also present the accuracy test for various physical parameters in Figure 2 with the same initial condition and $\alpha = 0$. This figure shows the $L^2$-errors as a function of the time step $\Delta t$, with variations in the physical parameters $\varepsilon$ and $M$. A second-order convergence rate is verified as the time step $\Delta t$ decreases. Additionally, we conducted numerical tests comparing our new Scheme 3 to the traditional second-order convex-splitting Scheme 4, which is based on the Crank–Nicolson formula. We fixed the parameters as follows: $N = 256$, $\Delta t = 0.001$, $T = 1$, $M = 0.02$, $\varepsilon = 0.05$, $\alpha = 0$, and $c_0 = 0.0$. The error of the traditional scheme was found to be $9.1 \times 10^{-5}$, while the error of our new scheme under the same parameters was significantly reduced to $1.9 \times 10^{-8}$. Both Schemes 3 and 4 are unconditionally stable. The computational time for both schemes with $\Delta t = 0.0001$ and $T = 1$ is quite close, taking 293 s and 261 s, respectively, to complete. This analysis offers an assessment of the numerical scheme's accuracy and its convergence behavior, providing valuable insights into its performance for simulating the diblock copolymer model.

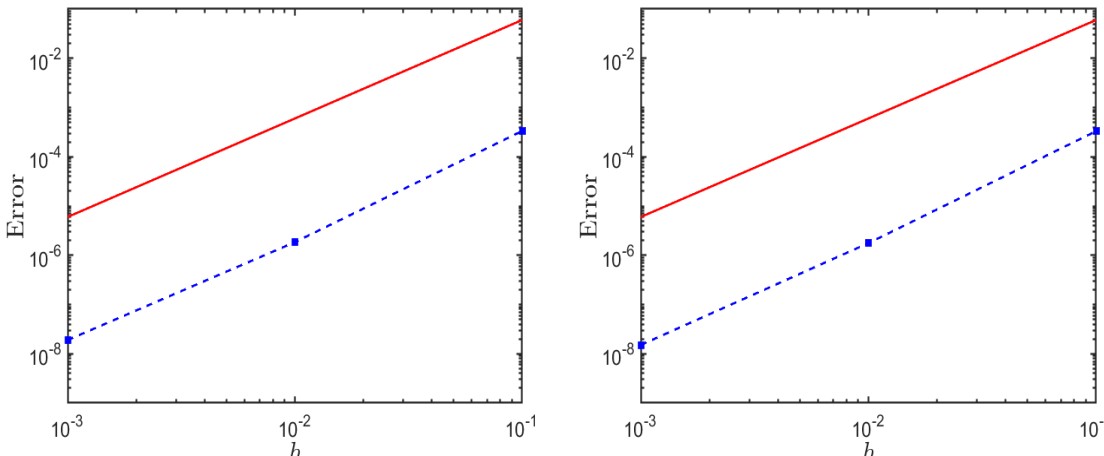

**Figure 1.** Time accuracy convergence test. The figures show the error versus time step $\Delta t$ for the diblock copolymer model using second-order Scheme 3. Two different initial parameters, $\alpha = 0.0$ (**left**) and $\alpha = 5.0$ (**right**), are used.

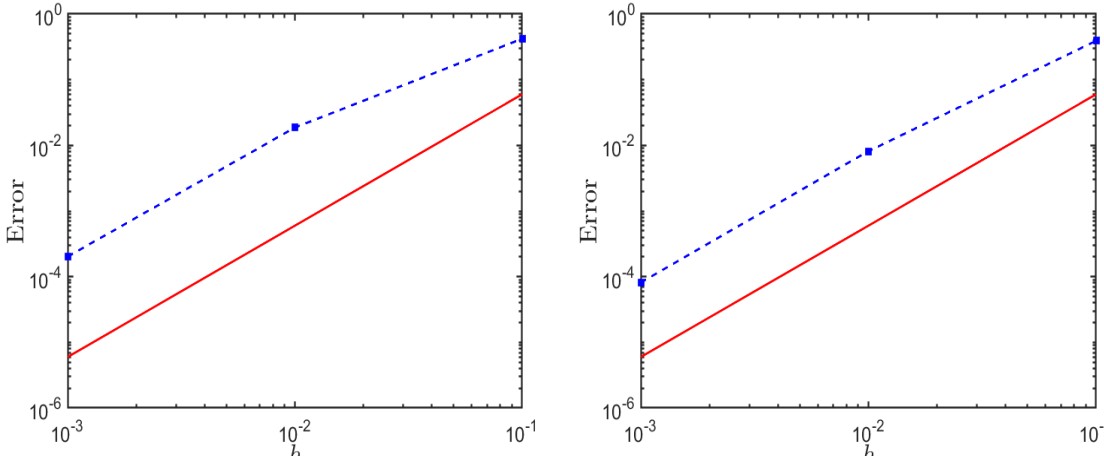

**Figure 2.** Time accuracy convergence test. The figures show the error versus time step $\Delta t$ for the diblock copolymer model using second-order Scheme 3. (**Left**): $M = 0.02$, $\varepsilon = 0.02$, $\alpha = 0$ and $c_0 = 0.0$; (**Right**): $M = 0.1$, $\varepsilon = 0.05$, $\alpha = 0$ and $c_0 = 0.0$.

### 4.2. Phase Separations

In this subsection, we present a simulation of phase separations using the diblock copolymer model (1). The physical and computational parameters are consistent with those used in the previous example. The focus of this simulation is to study the time evolution of the energy associated with the diblock copolymer model. The time interval considered is $t \in [0, 150]$. The plot of the time evolution of energy with different physical parameters, $c_0 = 0.0$ (left of the first row), $c_0 = 0.2$ (left of the first row), and $c_0 = 0.4$ (second row) in semi-log scale in Figure 3, allows to assess how well the numerical Scheme 3 performs under various conditions and choices of time step $\Delta t$.

Next, we will investigate the impact of different initial conditions on the coarsening dynamics of the diblock copolymer model. The simulations are conducted with a fixed time step of $\Delta t = 1.0^{-3}$. The numerical profiles of the phase variable $\phi$ are examined at various time points: $t = 1, 5, 10, 20, 40, 150$. The contour plots are presented with different values of the initial conditions in Figure 4 ($c_0 = 0.0$), Figure 5 ($c_0 = 0.2$), and Figure 6 ($c_0 = 0.4$). The results and observations for each case are summarized in the respective figures. Figure 4 suggests that the blue and red regions become entangled, leading to the formation of a cylindrical phase as the equilibrium solution. Figure 5 shows the coexistence of body-centered-cubic and cylindrical phases at $T = 150$, with the latter phase being dominant. Figure 6 indicates the presence of body-centered-cubic phases in the entire domain throughout the evolution process. In summary, this investigation explores how different initial conditions, represented by various values of $c_0$, influence the coarsening dynamics of the diblock copolymer model over time. The contour plots visually illustrate the evolution of the phase variable $\phi$ and provide insights into the formation of different phases and their dominance in the equilibrium state.

Finally, we plot the time evolution of energy decay for the diblock copolymer model under different sets of parameters. The time step of the simulation is fixed at $\Delta t = 10^{-3}$. For the first set of Figure 7, we set the parameter $\alpha$ to a constant value of 5, while $c_0$ varies. In the second set of Figure 8, we set $c_0$ to 0.4, and $\alpha$ varies. We observe how these parameters influence the rate of energy decay with different values of $c_0$ (Figure 7). Figure 8 shows how the energy decreases over time for each value of $\alpha$. It can be seen that the energy decreases at the same rate initially and decreases faster as $\alpha$ becomes smaller. The observations from these plots may reveal trends in the energy decay behavior as a function of the chosen parameters. This analysis helps in understanding the influence of parameter choices on the energetics of the diblock copolymer model, providing valuable insights into the model's behavior and stability.

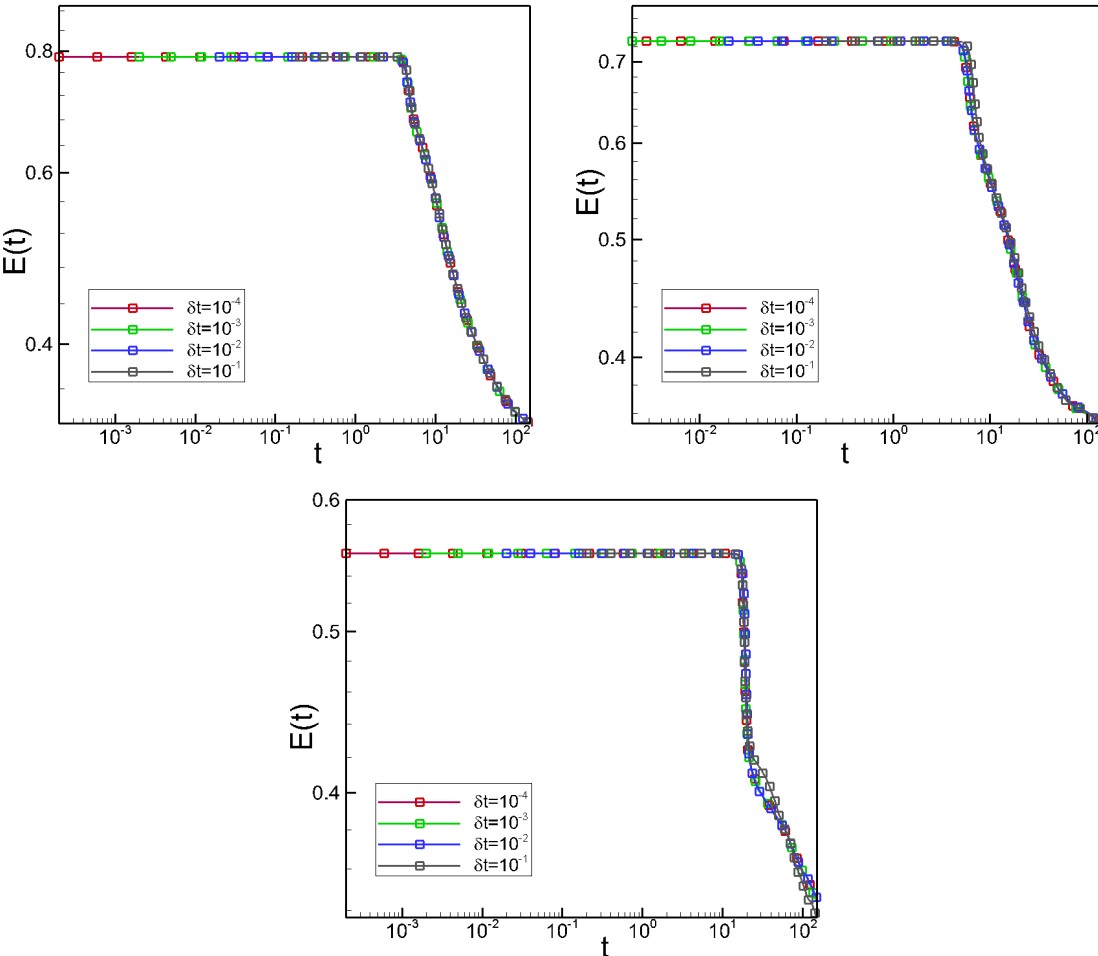

**Figure 3.** Time evolution of the energy decay for the diblock copolymer model. The time step is set at $\Delta t = 10^{-3}$ with different physical parameters $c_0 = 0.0$ (left of the first row), $c_0 = 0.2$ (right of the first row), and $c_0 = 0.4$ (second row).

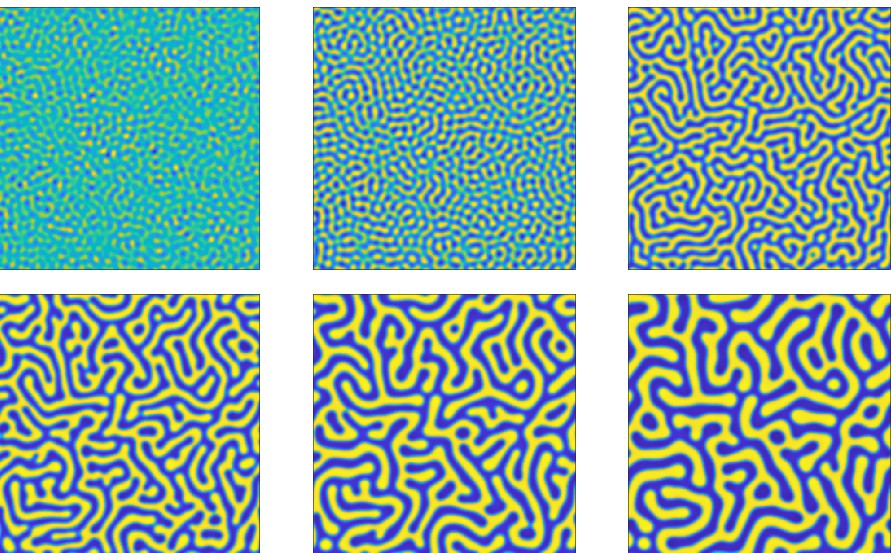

**Figure 4.** Time snapshots of coarsening dynamics driven by the phase field diblock copolymer model with random initial conditions. The parameters are set as $M = 0.02, \varepsilon = 0.05$. The profiles of $\phi$ at different time slots $t = 1, 5, 10, 20, 40, 150$, with $c_0 = 0$, are presented.

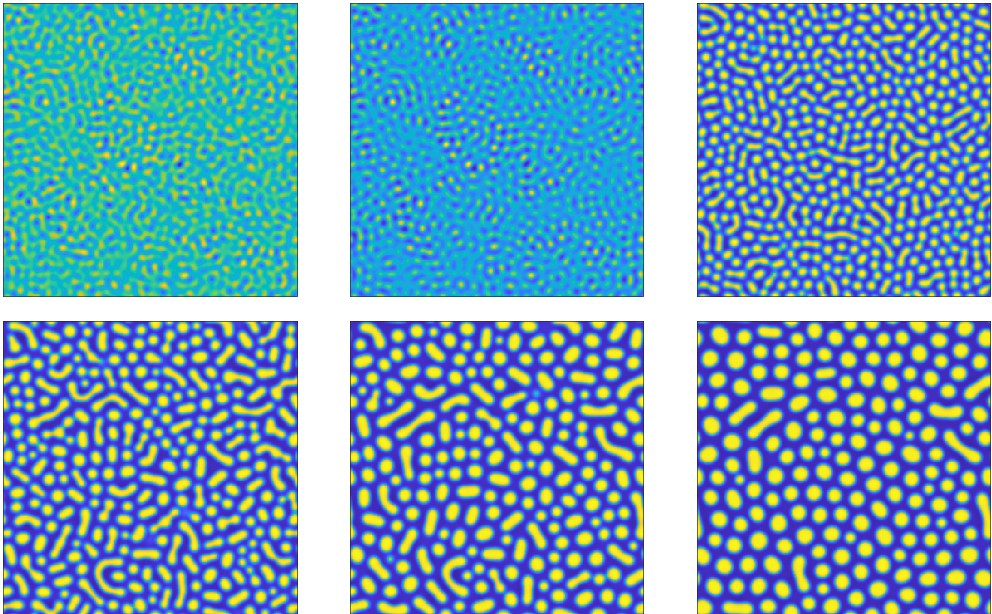

**Figure 5.** Time snapshots of coarsening dynamics driven by the phase field diblock copolymer model with random initial conditions. The parameters are set as $M = 0.02, \varepsilon = 0.05$. The profiles of $\phi$ at different time slots $t = 1, 5, 10, 20, 40, 150$, with $c_0 = 0.2$, are presented.

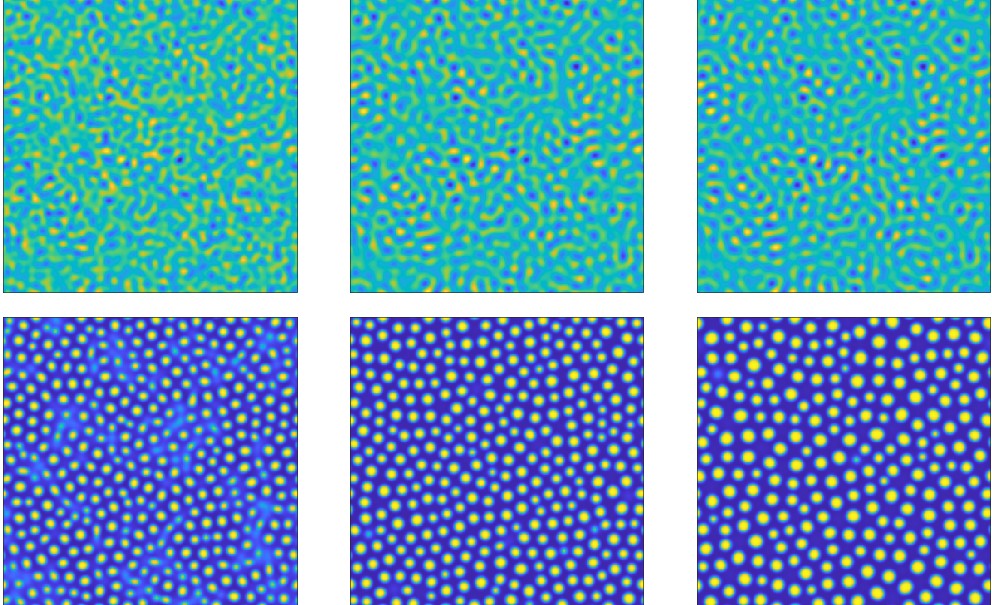

**Figure 6.** Time snapshots of coarsening dynamics driven by the phase field diblock copolymer model with random initial conditions. The parameters are set as $M = 0.02, \varepsilon = 0.05$. The profiles of $\phi$ at different time slots $t = 1, 5, 10, 20, 40, 150$, with $c_0 = 0.4$, are presented.

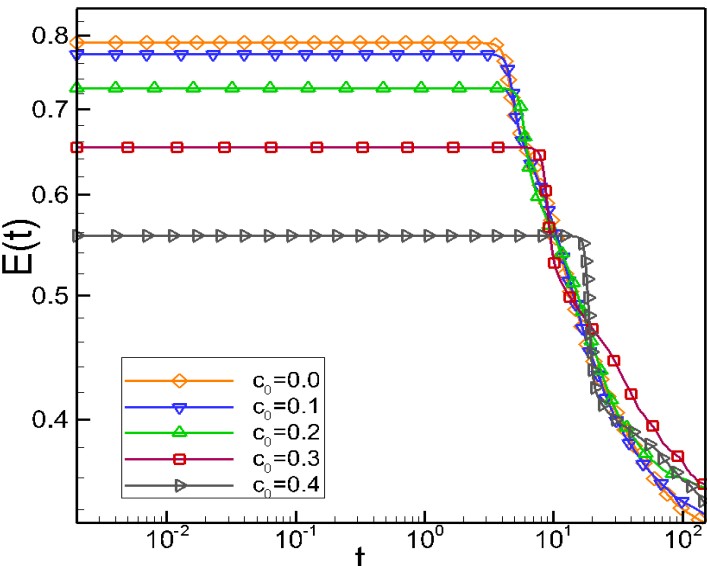

**Figure 7.** Time evolution of the energy decay for the diblock copolymer model. The time step is set at $\Delta t = 10^{-3}$, $\alpha = 5$ with different parameters $c_0 = 0.0, 0.1, 0.2, 0.3, 0.4$.

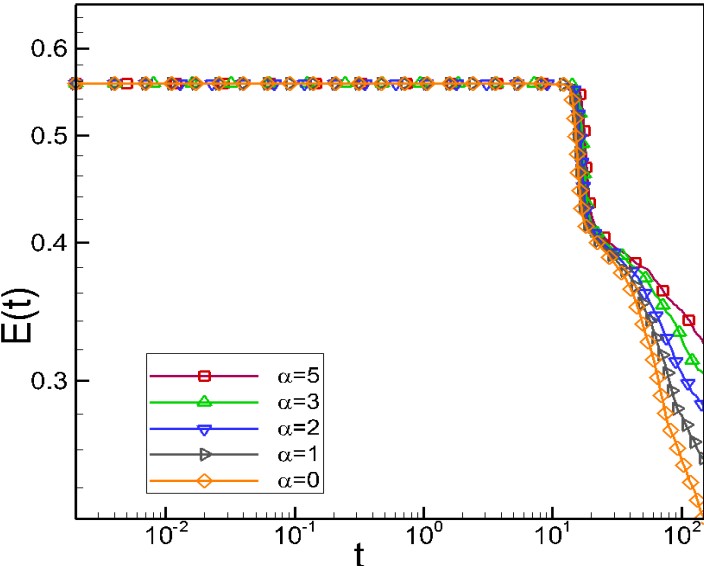

**Figure 8.** Time evolution of the energy decay for the diblock copolymer model. The time step is set at $\Delta t = 10^{-3}$ and $c_0 = 0.4$ with different $\alpha = 0, 1, 2, 3, 5$.

## 5. Conclusions

This paper focuses on introducing a second-order numerical scheme that utilizes the leapfrog time-marching method to approximate phase field models. The scheme offers several advantages and properties, which are highlighted in this paper. Firstly, the proposed numerical scheme achieves second-order accuracy in time. Secondly, the scheme is linear in nature, which simplifies the computational process and can lead to more efficient calculations. Thirdly, the proposed scheme preserves unconditional energy stability, which ensures that the numerical solutions generated by the scheme remain physically meaningful and do not exhibit unphysical behavior during simulations.

In addition to presenting the scheme's features, a series of tests and numerical examples are included to validate its effectiveness and reliability. These include convergence tests, where the scheme's convergence properties are examined, as well as numerical examples that demonstrate its applicability in solving phase field models. All the tests

mentioned in this paper are conducted using custom-developed code written in the Fortran programming language and executed on the Linux platform.

**Author Contributions:** Methodology, L.C. and Y.M.; software, L.C. and G.Z.; writing—original draft preparation, L.C. and G.Z.; writing—review and editing, Y.M., B.R. and G.Z. All authors have read and agreed to the published version of the manuscript.

**Funding:** This research was funded by National Natural Science Foundation of China Grants number 12071020, 12131005, and U2230402.

**Data Availability Statement:** Not applicable.

**Acknowledgments:** Lizhen Chen would like to acknowledge the support received from the National Natural Science Foundation of China, specifically Grants No. 12131005 and U2230402. The work of Bo Ren and Guohui Zhang were partially supported by the National Natural Science Foundation of China Grants No. 12071020, 12131005, and U2230402.

**Conflicts of Interest:** The authors declare no conflict of interest.

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
