# Peer review of "Numerical Approximations of Diblock Copolymer Model Using a Modified Leapfrog Time-Marching Scheme"

_computation, doi:10.3390/computation11110215_

Round 1
Reviewer 1 Report
Comments and Suggestions for Authors
The authors in this paper introduced an efficient modified leap-frog time-marching scheme for the diblock copolymer model and they demonstrated some advantages. They also displayed a numerical example in order to validate the proposed algorithm, showing its accuracy and efficiency in practical applications.
In my opinion, the paper is well-written and well-organized. But there are some minor amendments:
1- The survey about the related work is not enough. Please add some related and updated references that are relevant to your work.
2- There is a title of Section 2 .0.1. Numerical Schemes without any description. Please write a paragraph to give in brief overview of the plan of this section.
3- The punctuation in some places is missing. Please take this point into consideration. For example In Eq. 23.
4- Please mention in the concluding section, the software that is used to obtain the numerical results.
5-Although the paper is well-written, some minor typos were found. For example, in the first line of Section 3, The focus of this section, "the" should be removed. The end of equation 6 ends by a comma not by a dot. Please check typos and punctuation throughout the paper.
Comments on the Quality of English Language
There are only some typos and punctation to be adopted
Reviewer 2 Report
Comments and Suggestions for Authors
The work presents a novel numerical scheme based on a modified leap-frog time-marching scheme for the solution of a diblock copolymer model. The model features second order accuracy in time and high stability. The model is of potential interest to the journal but revision of some aspects is necessary:
- The authors must test the functioning of the numerical model in a wider range of parameters beyond initial condition. This is necessary to visualize the high stability claimed by the authors.
- The authors must include a comparison in terms of stability and computational time compared to previous numerical schemes used in the literature.
- The writing must be revised. There are typos throughout the manuscript. For example, in line 196 “The focus of the this section” should be “The focus of this section” and in line 216 “We computes” should be “We compute”.
Comments on the Quality of English LanguageMinor grammatical typos must be revised in the manuscript.
